# Tightly-orchestrated rearrangements govern catalytic center assembly of the ribosome

Yi Zhou [1], Sharmishtha Musalgaonkar[1], Arlen W. Johnson[1,2] & David W. Taylor [1,2,3,4]

The catalytic activity of the ribosome is mediated by RNA, yet proteins are essential for the function of the peptidyl transferase center (PTC). In eukaryotes, final assembly of the PTC occurs in the cytoplasm by insertion of the ribosomal protein Rpl10 (uL16). We determine structures of six intermediates in late nuclear and cytoplasmic maturation of the large subunit that reveal a tightly-choreographed sequence of protein and RNA rearrangements controlling the insertion of Rpl10. We also determine the structure of the biogenesis factor Yvh1 and show how it promotes assembly of the P stalk, a critical element for recruitment of GTPases that drive translation. Together, our structures provide a blueprint for final assembly of a functional ribosome.

[1] Department of Molecular Biosciences, University of Texas at Austin, Austin, TX 78712, USA. [2] Institute for Cellular and Molecular Biology, University of Texas at Austin, Austin, TX 78712, USA. [3] Center for Systems and Synthetic Biology, University of Texas at Austin, Austin, TX 78712, USA. [4] LIVESTRONG Cancer Institutes, Dell Medical School, Austin, TX 78712, USA. Correspondence and requests for materials should be addressed to A.W.J. (email: arlen@austin.utexas.edu) or to D.W.T. (email: dtaylor@utexas.edu)

Ribosomes are the molecular machines that all cells depend on for protein synthesis. Its two fundamental functions, decoding messenger RNAs and polypeptide synthesis, are separated into the small subunit and large subunits, respectively. Despite using RNA for catalysis, ribosomes are ribonucleoprotein particles, and proteins surrounding the peptidyl transferase center (PTC) are essential for function. In eukaryotes, the ribosomal subunits are largely preassembled in the nucleolus where the ribosomal RNAs are transcribed[1–5]. However, ribosomal subunits are exported to the cytoplasm in a functionally inactive and immature state, requiring the further addition of ribosomal proteins and the removal of transacting factors that block ligand binding sites[6–9]. As a consequence, the assembly of ribosomes is coupled to their nuclear export.

In budding yeast, nuclear export of nascent pre-60S subunits requires the export adapter Nmd3[10,11], the mRNA export factor Mex67-Mtr2[12], the degenerate methionyl amino peptidase Arx1[13,14], and several other proteins reviewed in refs. [15,16]. However, only Nmd3 appears to have a universally conserved role as an export factor in eukaryotes. Interestingly, Nmd3 homologs are also found in archaea, suggesting that the protein has a function in ribosome assembly that predates the evolution of the nuclear envelope and its role as an export factor. Nmd3 is a multidomain protein that we and others previously showed spans the entire joining face of the 60S subunit[17,18]. Its eIF5A domain occupies the E site, while additional domains bind in the P site and occlude the A site, rendering the joining face inaccessible to transfer RNAs and other large subunit ligands. A small entourage of additional biogenesis factors accompanies the pre-60S to the cytoplasm reviewed in ref. [15]. Among these factors, Tif6 blocks association with the small subunit[19,20] to prevent premature engagement of the assembling 60S.

In the cytoplasm, the pre-60S particle follows a hierarchical pathway of assembly events coordinated with the release of biogenesis factors[21]. Cytoplasmic maturation is initiated by the AAA-ATPase Drg1, which is recruited to the subunit and activated via Rlp24[22], a paralog of the ribosomal protein Rpl24. Release of Rlp24 appears to be coordinated with the release of the GTPase Nog1[23], which disrupts the A site while its C-terminal extension is inserted into the polypeptide exit tunnel[24]. Downstream completion of the subunit requires assembly of the P (L7/L12) stalk, which recruits and activates the GTPases of the translation cycle[25], and insertion of Rpl10 (uL16), to complete the PTC. Molecular genetics analyses showed that assembly of the P stalk requires the dual-specificity phosphatase Yvh1 to release the placeholder protein Mrt4, a paralog of the P stalk protein P0 (uL10)[26,27]. Similarly, functional interactions among *RPL10*, *NMD3* and *LSG1*, encoding a second GTPase, suggest an interplay among these factors in promoting the loading of Rpl10 and the release of Nmd3[28–32]. The insertion of Rpl10 into the pre-60S particle completes the subunit, priming it for quality control. The integrity of the PTC is subsequently assessed in a test drive[33,34] which uses molecular mimics of translation factors[34,35] to license the subunit for translation by the release of Tif6. Defects in this test drive are associated with Shwachman–Diamond syndrome in humans[36,37]. While extensive molecular genetics and biochemical studies over the past 20 years have provided a framework for understanding the complex process of cytoplasmic maturation, the mechanisms for assembly of both the P stalk and the PTC have remained elusive.

Recent advances in cryo-electron microscopy (cryo-EM) give us the ability to resolve intermediates of ribosome assembly at near atomic resolution. Multiple structures of nucleolar and nuclear pre-60S intermediates have been reported[24,38–41] (reviewed in refs. [42–45]). However, only a single high-resolution structure of a native cytoplasmic intermediate has been reported[18]. Here, we have used cryo-EM to determine the structures of a series of intermediates of 60S maturation that reveal the dynamic changes in RNA conformations and protein exchanges required for assembly of the P stalk and completion of the PTC.

## Results

### Prying open of RNA helices primes the 60S for Rpl10 loading.
Using cryo-EM of 60S precursors trapped from yeast, we captured six intermediates in late nuclear and cytoplasmic 60S assembly at ~3.5–3.8 Å resolution (Fig.1, Supplementary Figs 1–4 and Supplementary Movie 1, 2), which provide unprecedented mechanistic insights into 60S maturation. To isolate particles immediately after export from the nucleus, we used tandem affinity purification (TAP)-tagged mutant Rlp24 (Rlp24ΔC), which fails to recruit the AAA-ATPase Drg1 that initiates cytoplasmic maturation[22]. Surprisingly, we identified both late nuclear (LN) and early cytoplasmic (EC) particles, the latest nuclear and earliest cytoplasmic particles visualized to date, respectively. The pre-export LN particle lacks the nuclear export adapter Nmd3 but contains the biogenesis factors Rlp24, Bud20, Mrt4, Nog1, Nsa2, Nog2, Tif6, and Arx1 (Fig. 1a). We also observe clear density for Rpl12 (uL11) on the P stalk. The L1 stalk, which interacts with E site ligands in a mature ribosome, is in an open conformation and H89, which constitutes part of the binding cleft for Rpl10, is displaced by the N-terminal domain (NTD) of Nog1 in the A site. In addition, H69, which provides a critical inter-subunit bridge in the complete 80S ribosome, is disrupted by Nog2. Earlier studies of other nuclear pre-60S intermediates showed that in both the Nog2 and Arx1 particles[24,40], the 5S has not yet rotated into its mature position and internal transcribed spacer 2 (ITS2) has not been processed. In the LN structure, the 5S RNA has rotated into its mature position and ITS2 has been removed, placing the LN particle later than the Arx1 and Nog2 particles in maturation. The presence of Rpl42 (eL42), which completes the binding site for Nmd3[17,18], and Rpl29 (eL29) in the LN particle also place it later in the assembly pathway than the previously described Rix1 particle[46]. Thus, the LN particle represents a pre-60S particle immediately before gaining export competence by exchanging Nog2 with Nmd3[47] to drive export to the cytoplasm.

In the progression from the LN to EC particles, Nmd3 has exchanged with Nog2 for export from the nucleus, and Nsa2 has been released (Fig. 1b). Nmd3 binding in the E and P sites promotes two large-scale rearrangements of RNA: closure of the L1 stalk and capture of H38, which constitutes the second RNA helix of the Rpl10 binding cleft. The L1 stalk is held closed by interactions between Rpl1 (uL1) and the eIF5A-like domain of Nmd3 in the E site. H38 fits snugly into a saddle-shaped surface of Nmd3 where it is stabilized by multiple contacts including Arg333 stacking on the flipped-out A1025 extending from the tip of H38. Strikingly, in this state, the tip of H38 is shifted ~50 Å from its position in the mature subunit (Supplementary Fig. 5a). Because H38 provides extensive contacts for Rpl10 in the mature subunit, this bending of H38 by Nmd3 partially opens the Rpl10 binding site to initiate priming of the pre-60S particle for loading of Rpl10.

The EC particles could be further separated into two states: early cytoplasmic-immediate (ECI) and early cytoplasmic-late (ECL). In the ECI particle, the NTD of Nog1 remains inserted in the A site, displacing H89 (Supplementary Fig. 5b) and preventing Rpl10 insertion into this particle. In addition, the ECI particle is the earliest particle in which H69 is folded into its nearly mature conformation. As this coincides with Nmd3 binding, Nmd3 may promote H69 folding. However, Nmd3 also

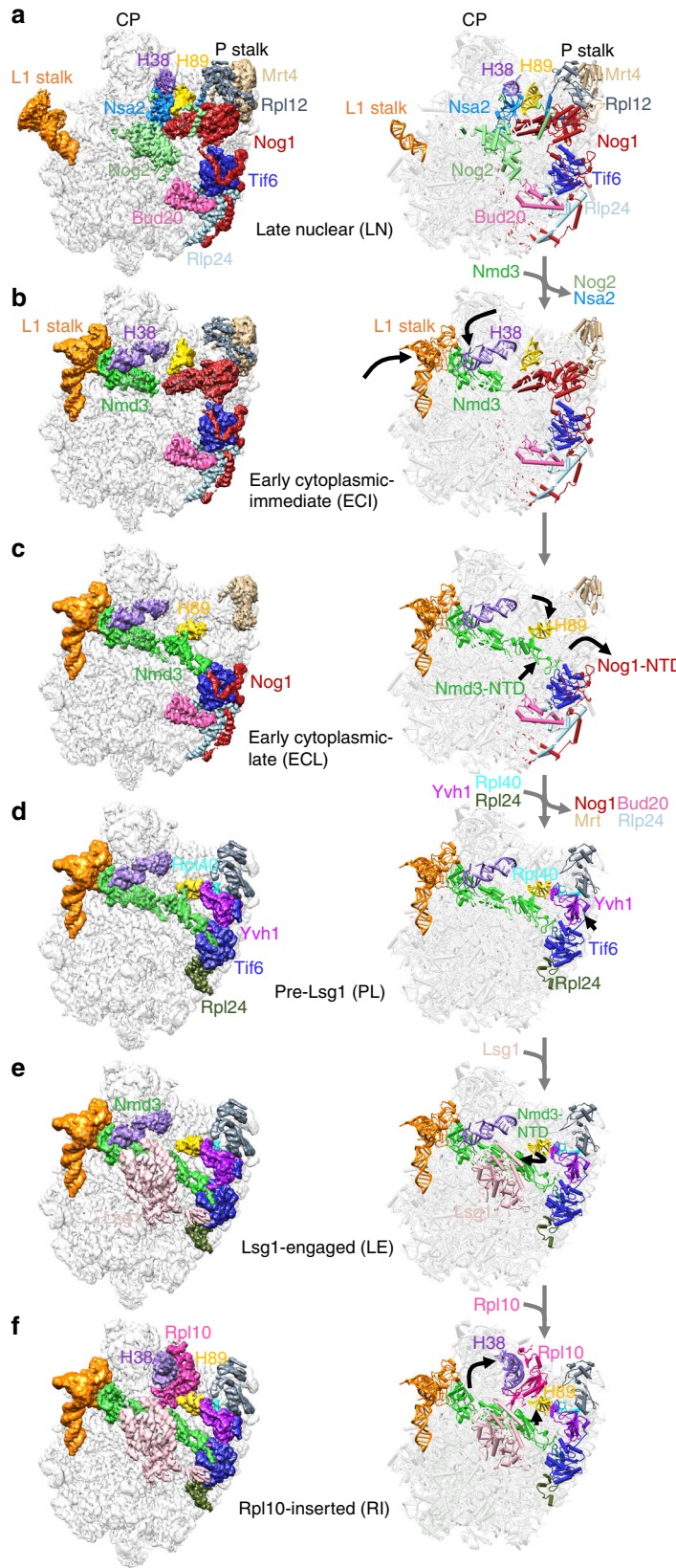

appears to drive the flipping of two bases, G2261 and U2269 (Supplementary Fig. 5c), distinguishing this structure from mature H69 and priming the particle for activation of the GTPase Lsg1, the release factor for Nmd3[17,28]. Interestingly, in the ECI particle, there is no discernible density for the NTD of

Nmd3 (Fig. 1b), most likely because it is highly mobile before docking onto Tif6.

In the ECL particle the N terminus of Nog1 has been displaced from the A site, allowing the NTD of Nmd3 to dock onto Tif6 (Fig. 1c). H89 now rearranges into its nearly mature position

**Fig. 1** Structures of late nuclear and cytoplasmic particles reveal the pathway of 60S maturation. Crown view of late nuclear (LN) (**a**), early cytoplasmic-immediate (ECI) (**b**), early cytoplasmic-late (ECL) (**c**), pre-Lsg1 (PL) (**d**), Lsg1-engaged (LE) (**e**), and Rpl10 (uL16)-inserted (RI) (**f**) ribosomal intermediates at ~3.5–3.8 Å resolution, with corresponding atomic models shown on the right. Key protein and ribosomal RNA (rRNA) elements are colored as indicated. Black arrows indicate conformational changes while gray arrows indicate protein exchanges. CP, central protuberance. In the early cytoplasmic-immediate (ECI) particle, Rpl12 (uL11) was not modeled due to poor density. In the early cytoplasmic-late (ECL) particle, Rpl12 is not clearly discernible and Mrt4 is poorly resolved due to the mobility of the P stalk after the N-terminal domain (NTD) of Nog1 is released from the A site. Therefore, Rpl12 was not modeled, and Mrt4 was only partially modeled

where it engages with a small loop within the NTD of Nmd3, which we name the histidine thumb (Supplementary Fig. 5b and Supplementary Movie 3). Comparing the ECI and ECL particles shows that release of Nog1 is necessary to allow Nmd3 docking (Supplementary Fig. 5d). Surprisingly, the C terminus of Nog1 remains in place on the ECL particle, intertwining with Rlp24 and occupying the exit tunnel (Fig. 1c). The persistence of the C terminus of Nog1 in the exit tunnel likely results from the failure to recruit Drg1 to the Rlp24ΔC particle and suggests that events on the joining face can be uncoupled from extraction of Nog1 from the exit tunnel by Drg1. Together, the interaction of the histidine thumb of Nmd3 with H89 and the capture of H38 by the eIF5A-like domain of Nmd3 hold open the Rpl10 binding site, priming the pre-60S for the loading of Rpl10.

To capture a series of particles downstream of the ECL particle, we partially arrested ribosome biogenesis in vivo with diazaborine, an inhibitor of the AAA-ATPase activity of Drg1[48]. The three classes of particles we obtained differ in their occupancy of Rpl10 and Lsg1 and show a large-scale structural rearrangement in Nmd3. In all three classes, Nog1, Bud20, Mrt4, and Rlp24 have been released, while Rpl40 (eL40) and Rpl24 (eL24) have loaded. The pre-Lsg1 (PL) particle contains Nmd3 with the NTD interacting with H89 similarly to the ECL particle; the Lsg1-engaged (LE) particle contains Nmd3 and Lsg1 with the NTD of Nmd3 rotated to engage Lsg1; and the Rpl10-inserted (RI) particle contains Nmd3, Lsg1 and Rpl10 with the NTD of Nmd3 remaining engaged with Lsg1 (Fig. 1d–f). Intriguingly, in all classes we observed a well-defined density between Tif6 and the P stalk (Supplementary Fig. 6a) that we identified as the zinc-binding domain (ZBD) of Yvh1. Yvh1 has 364 amino acids, comprising an N-terminal phosphatase domain (amino acids (aa) 1–214) and a C-terminal ZBD (aa 215–364). While the function of the phosphatase domain is unknown, the ZBD promotes Mrt4 release for P stalk assembly[26,27].

**Yvh1 releases Mrt4 by rearranging the P stalk**. We were able to build an atomic model of the ZBD of Yvh1 ab initio (Supplementary Fig. 6). The ZBD of Yvh1 is wedged between the P stalk and Tif6 and centered on the sarcin-ricin loop (SRL), an RNA element that is essential for activating translational GTPases. An extended internal loop of Yvh1 also engages the tip of H89 (Fig. 2a). The ZBD is composed almost entirely of beta-sheets and contains two predicted zinc-binding centers (Supplementary Fig. 6c–e). Yvh1 interacts directly with the P stalk, H89, and the SRL, with Trp329 stacking on G1242 of H43, R269 stacking on C2284 of H89, and Phe260 stacking on A3027 of SRL (Fig. 2b–d). An additional density adjacent to the P stalk that may account for the N-terminal phosphatase domain of Yvh1 can be seen at lower thresholds (Supplementary Fig. 7a). The C terminus of Yvh1 was less well resolved but could be traced to the surface of Tif6 at lower thresholds (Supplementary Fig. 7b), where it interacts with the extreme N terminus of Nmd3. Interestingly, the C-terminal 20 amino-acid tail of Tif6 folds back over itself to interact with Nog1 in the Nog1-bound particle, but reorients toward Yvh1 in the Yvh1-bound particle (Supplementary Fig. 7c). Comparison of the Nog1-containing ECI particle with the Yvh1-bound PL

particle reveals that the NTD of Nog1 (Supplementary Fig. 7d) and the displacement of H89 by Nog1 (Supplementary Fig. 5b) would both occlude Yvh1 binding. Thus, Nog1 controls the timing of Yvh1 binding after export to the cytoplasm.

To understand how Yvh1 releases Mrt4 for P stalk assembly, we compared the P stalk from Mrt4-containing ECI particles and Yvh1-containing PL particles. Intriguingly, we find that Yvh1 binding leads to allosteric changes in the binding site of Mrt4 to release this protein. The LN particle along with the ECI and ECL particles contain Mrt4 on the P stalk, whereas the PL particle lacks Mrt4 but contains Yvh1, consistent with previous work showing that the binding of Yvh1 displaces Mrt4[26,27]. Notably, in the PL particle, the P stalk has undergone a rotation of ~20° away from the central protuberance toward Tif6 (Fig. 2e). This rotation involves a rigid body rotation of the tip of the stalk (H43 and 44) with Rpl12 that is accomplished by bending of H42, which forms the stem of the stalk. The docking site of Mrt4 includes H43 and H44 as well as A1221 of H42, which inserts into a pocket in Mrt4 (Fig. 2f). Aligning H43, H44, and Rpl12 from the Mrt4-bound and Yvh1-bound structures reveals that the Yvh1-induced bending of the P stalk RNA retracts the Mrt4 pocket away from A1221 on H42 (Fig. 2g, h), thereby reducing Mrt4 contacts with RNA. Mutations in Mrt4 that reduce its affinity for RNA bypass the need for Yvh1[26,27]. These mutations include residues K23 and K69 that interact with A1221 (Fig. 2f), demonstrating that the interaction of Mrt4 with this RNA element is critical for its stable binding to the P stalk. Thus, Yvh1 controls the release of Mrt4 by allosterically altering its binding site, allowing final assembly of the P stalk proteins and setting up the subunit for a GTPase-dependent test drive.

**A large rotation of Nmd3 releases H89 for Rpl10 loading**. In the transition from the ECL particle to the PL particle, the NTD of Nmd3 remains docked on Tif6 (Fig. 1d). The NTD of Nmd3 contains two zinc-binding centers (Fig. 3a). Residues 16–39 form a compact treble clef that binds to Tif6 with the N-terminal residues of Nmd3 extending over the surface of Tif6. The C-terminal end of this zinc center contains a short alpha helix that connects to the remainder of the N-terminal domain through a single loop of amino acids. This is mirrored on the distal end of the NTD, where the second zinc-binding treble clef, comprised of C56–C58–C143–C145, connects to the eL22-like domain of Nmd3 in the P site through another loop of amino acids. Thus, the bulk of the NTD of Nmd3 is suspended by loops on each end, allowing this domain to pivot.

In the LE particle, Lsg1 interacts extensively with H69 with Trp142 stacking on A2256 at the tip of the helix and Phe374 stacking on U2260. The two flipped bases of H69, G2261 and U2269, interact with a strand of Lsg1 adjacent to switch 1 of its GTPase center and Asp295 of Nmd3, respectively (Fig. 3b). Remarkably, the NTD of Nmd3 undergoes an ~60° rotation upon Lsg1 binding (Figs. 1e and 3c, d). In this rotated state, the histidine thumb has swung away from H89 to engage with the long alpha helix (aa 149–173) of Lsg1. This rotation involves swiveling of the N-terminal domain about the two linkers that connect this domain on one end to the zinc center,

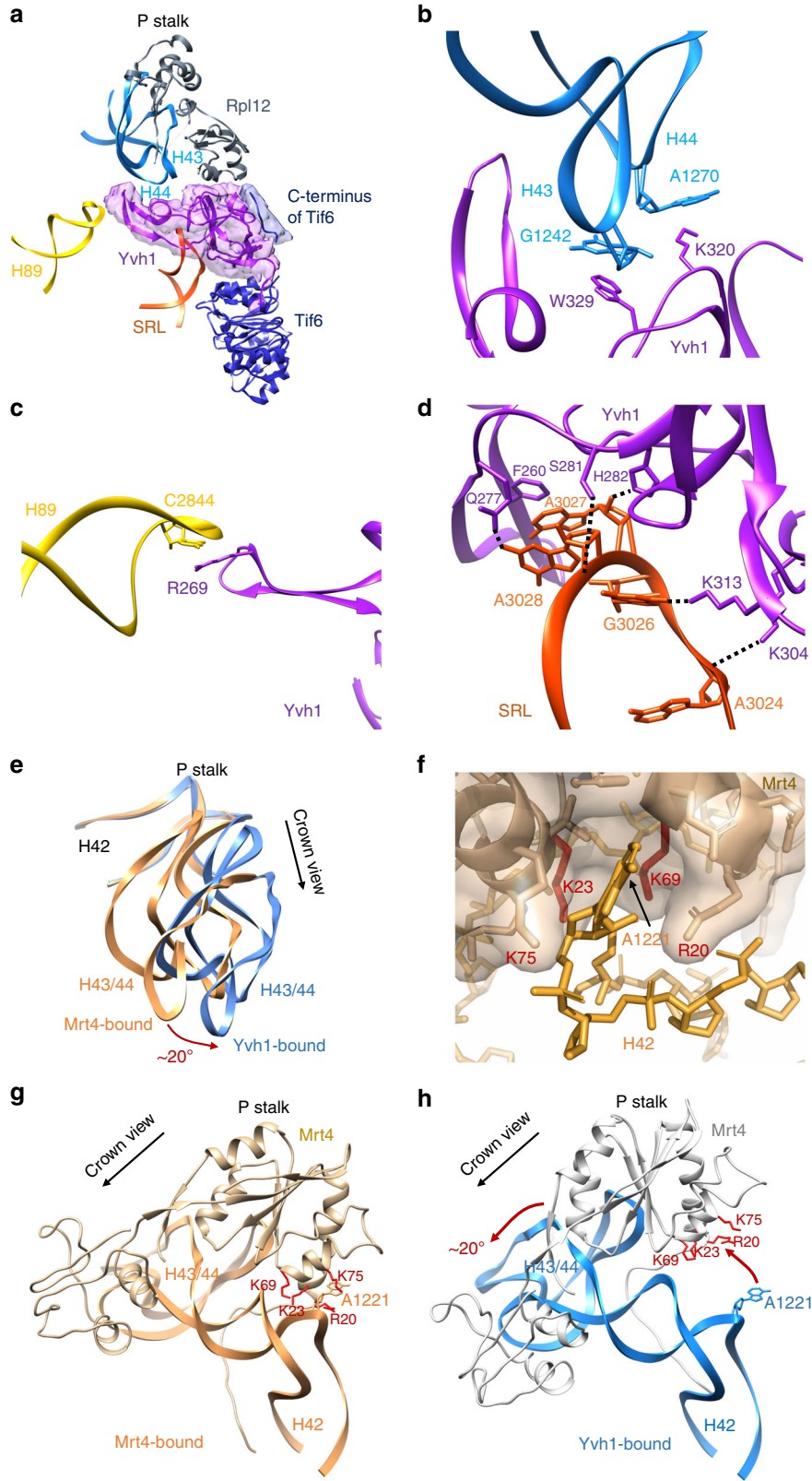

**Fig. 2** Yvh1 forces the release of Mrt4. **a** Atomic structure of the zinc-binding domain of Yvh1 modeled into the unassigned density. The small piece of extra density (blue) was assigned to the C terminus of Tif6 (see Supplementary Fig. 7b). **b–d** Interactions of Yvh1 with helices H43 and H44 (**b**), helix H89 (**c**), and sarcin-ricin loop (SRL) (**d**). Dashed lines, predicted hydrogen bonds. **e** Comparison of the P stalk RNAs in the Mrt4-bound (orange) and Yvh1-bound (blue) states. **f** Mrt4 in cartoon and transparent surface representation. A1221 of H42 bound in a pocket of Mrt4. **g** Mrt4 bound to P stalk RNA rotated ~120° clockwise relative to the view in (**a**) showing A1221 of H42 in the binding pocket of Mrt4. Residues of Mrt4 interacting with H42 are indicated. **h** Mrt4 from the early cytoplasmic-immediate (ECI) particle was docked onto H43 and H44 of the Yvh1-bound pre-Lsg1 (PL) particle, showing the movement of Mrt4 away from A1221 of H42

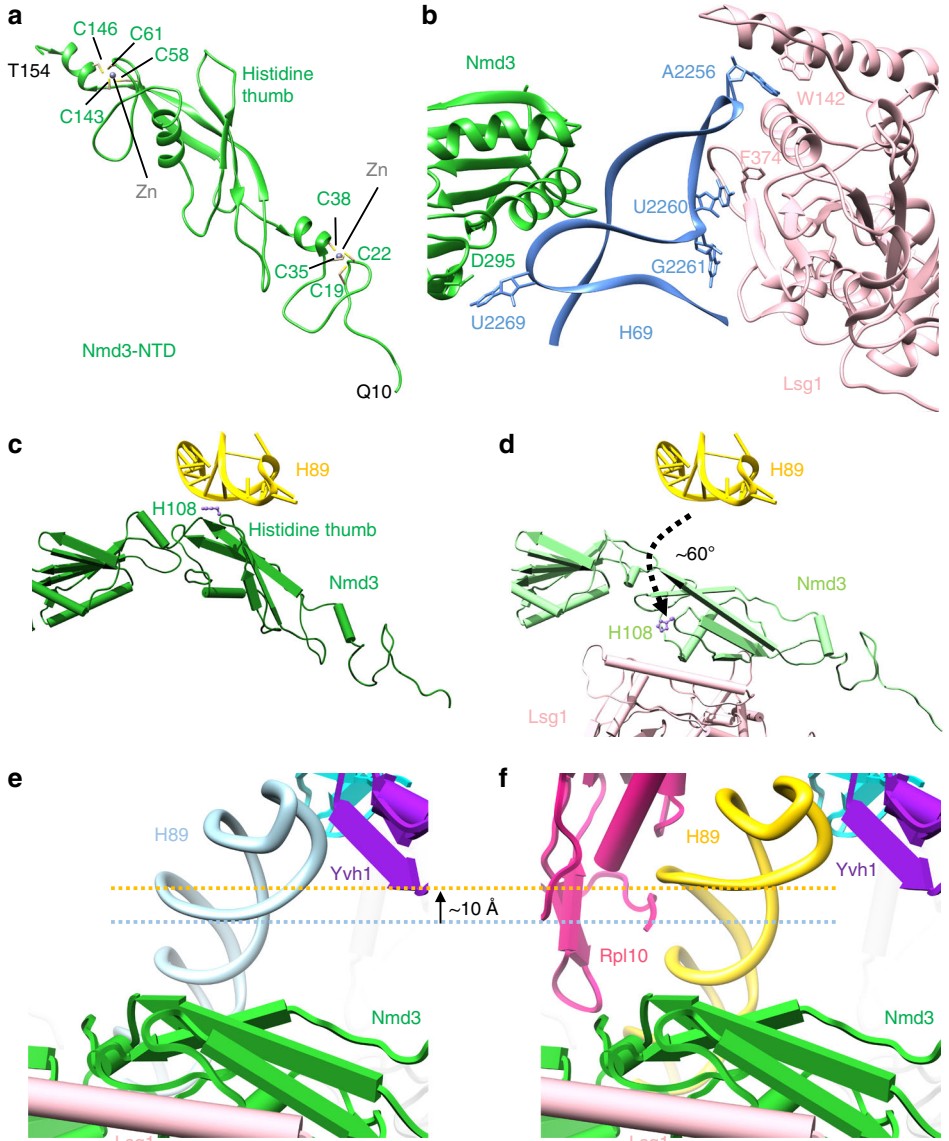

**Fig. 3** Large-scale rearrangements of Nmd3 N-terminal domain (NTD) primes the subunit for Rpl10 (uL16) insertion and completion of the catalytic center. **a** Atomic structure of the zinc-binding NTD of Nmd3. Cysteines of the two treble clef zinc-binding motifs are indicated. Zn$^{2+}$ ions were modeled into their predicted positions. **b** Interactions of H69 with Nmd3 and Lsg1 showing flipped-out G2261 and U2269. **c** In the pre-Lsg1 particle, H108 of the Nmd3 histidine thumb is engaged with H89. **d** Upon Lsg1 binding, the bulk of the NTD of Nmd3 rotates ~60°, releasing the histidine thumb from H89. **e**, **f** Upon Rpl10 insertion, the middle portion of H89 is retracted ~10 Å to accommodate and stabilize Rpl10 in its binding cleft; the tip of H89 remains in position to interact with Yvh1

docked on Tif6, and on the other end, to the eL22-like domain in the P site. This rotation is also accompanied by a slight displacement of the C-terminal end of the zinc-binding domain of Nmd3 toward Lsg1 (Supplementary Movie 4). Importantly, the release of H89 from the histidine thumb of Nmd3 frees this helix to adopt its final, mature position upon Rpl10 insertion. Therefore, the LE particle is primed for Rpl10 loading by both the prying open of H38 and release of H89 by Nmd3.

**Release of H38 and H89 from Nmd3 stabilizes Rpl10.** Ultimately, insertion of Rpl10 promotes two RNA conformational changes in the Rpl10-inserted (RI) particle. H38 is retracted away from the eIF5A domain of Nmd3 to adopt its mature position, where it makes extensive contacts with Rpl10 (compare Fig. 1f with Fig. 1e, Supplementary Movie 5). Additionally, there is a

subtle (~10 Å) yet important retraction of the middle portion of H89 towards Rpl10, which drives H89 into its mature position to stabilize Rpl10 in its binding cleft between H38 and H89 (Fig. 3e, f, Supplementary Movie 6). Unexpectedly, Nmd3 remains in place in progression from the LE to the RI particle (Fig. 1f), suggesting that the insertion of Rpl10, alone, is not sufficient to displace Nmd3 as previously proposed[18].

To test the model that Nmd3 holds open RNA helices to promote Rpl10 loading, we mutated residues in Nmd3 that directly contact H38, based on the PL structure (Fig. 4a). We then asked if these mutants could suppress *rpl10-G161D*, a temperature-sensitive mutation that destabilizes Rpl10 binding to the ribosome. All structure-guided mutations, including mutation of Arg333, which interacts with the flipped-out base A1025 at the tip of H38, suppressed *rpl10-G161D* to varying degrees (Fig. 4c). Additionally, mutations in Nmd3 that

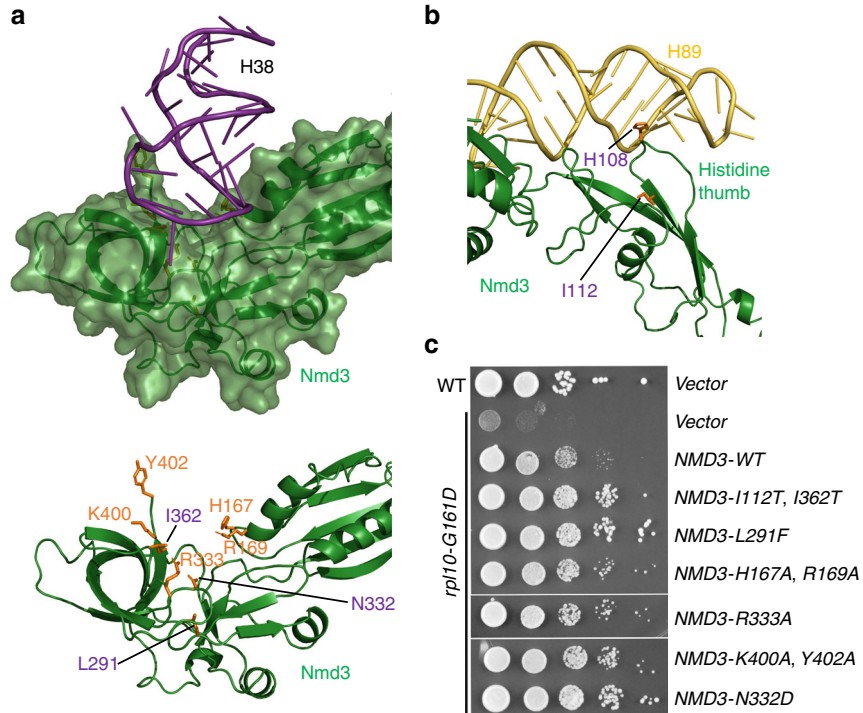

**Fig. 4** Release of H38 and H89 from Nmd3 stabilizes Rpl10 (uL16) in the ribosome. **a** Atomic structure showing that H38 lays in a saddle of Nmd3 (top). Lower panel, selected residues highlighted in orange sit in the immediate interface between Nmd3 and H38. L291, N332, and I362 (purple) were previously identified from genetic screens for suppressors the temperature-sensitive *rpl10-G161D* mutant. **b** Atomic structure showing that interaction of the histidine thumb of Nmd3 with H89. H108 and I112 were identified from genetic screens *rpl10-G161D* suppressors. **c** Wild-type (WT) or *rpl10-G161D* mutant cells were transformed with empty vector or vector expressing WT or the indicated NMD3 mutants. Ten-fold serial dilutions of cultures were plated onto selective plates and incubated for 2 days at 30 °C, a semi-permissive temperature for *rpl10-G161D*. The mutations H167A, R169A, R333A, K400A, and Y402A in *NMD3* were engineered based on the structural information as indicated in (**a**)

suppress *rpl10-G161D*, which had been identified in early genetic screens[30,49], map to the saddle of the eIF5A-like domain that interacts with H38 and to the histidine thumb of Nmd3 that interacts with H89 in our structure (Fig. 4a, b). These mutations were highly specific to *rpl10-G161D* as they did not suppress a different mutation in *RPL10*, *rpl10-R98S*, that blocks Nmd3 release after Rpl10 insertion (Supplementary Fig. 8)[31]. Importantly, these suppressing mutations weaken the affinity of Nmd3 to the 60S[28], which we can now unambiguously attribute to weakened binding to H38 and H89. Taken together, these results suggest that the release of H38 and H89 from Nmd3 stabilizes Rpl10 in its binding cleft. Thus, the export adapter Nmd3 plays a critical role in both priming the binding site for Rpl10 loading and stabilizing Rpl10 in the ribosome to complete the PTC.

## Discussion

Here, we have used cryo-EM to determine the structures of six different intermediates of 60S maturation, ranging from a late nuclear species to the completion of the PTC. We can now propose a comprehensive, detailed model for catalytic center assembly of the ribosome (Fig. 5). This work builds on our earlier genetic analysis of 60S maturation, where we showed that cytoplasmic events could be ordered into a hierarchical pathway[21]. Cytoplasmic maturation is initiated by the AAA-ATPase Drg1, which removes the placeholder protein Rlp24. Subsequently, the maturation pathway bifurcates into events centered at both the exit tunnel and on the joining face. On the joining face, maturation culminates in the assembly of the catalytic center and the P stalk, two elements that are critical for the function of the

ribosome. Completion of events in both branches of the pathway is necessary for the subsequent functional check of the subunit and its licensing for translation. In this work, we focused on the events on the joining face to understand assembly of the PTC and the P stalk.

Nuclear export of the 60S subunit requires the conserved multidomain export adapter Nmd3[10,11]. In the late nuclear particle, the loading of Nmd3 is blocked by the GTPase Nog2 (Fig. 5a)[47]. Although we do not know how Nog2 is activated or released from the pre-60S particle, our LN structure reveals that the biogenesis factor Rfp2 has been removed at this stage in assembly. The C terminus of Rpf2 interdigitates between Nog2 and H69 and stabilizes a loop of Nog2 through a beta-sheet interaction. Removal of Rpf2 appears to destabilize H69 as we were unable to resolve this helix in the LN structure. These observations lead us to speculate that the removal of Rpf2 allows conformational changes in Nog2 and H69 that promote the activation of its GTPase activity and/or its release to allow Nmd3 to bind.

Upon initial binding of Nmd3 (Fig. 5b), its eIF5A- and eL22-like domains occupy the E and P sites, respectively, where they close the L1 stalk and capture H38. In this state, the N terminus of Nmd3 is sterically blocked from docking on Tif6 by the presence of the N terminal and GTPase domains of Nog1, which disrupt H89 in the A site. The C terminus of Nog1, on the other hand, wraps around the back of the subunit and enters the exit tunnel. The release of Nog1 is thought to be coupled to the Drg1-dependent release of Rlp24, presumably by extracting the C terminus of Nog1 from the exit tunnel[23,48]. However, in the ECL particle, recruitment of Drg1 is blocked due to truncation of the C terminus of Rlp24. Consequently, the C terminus of Nog1

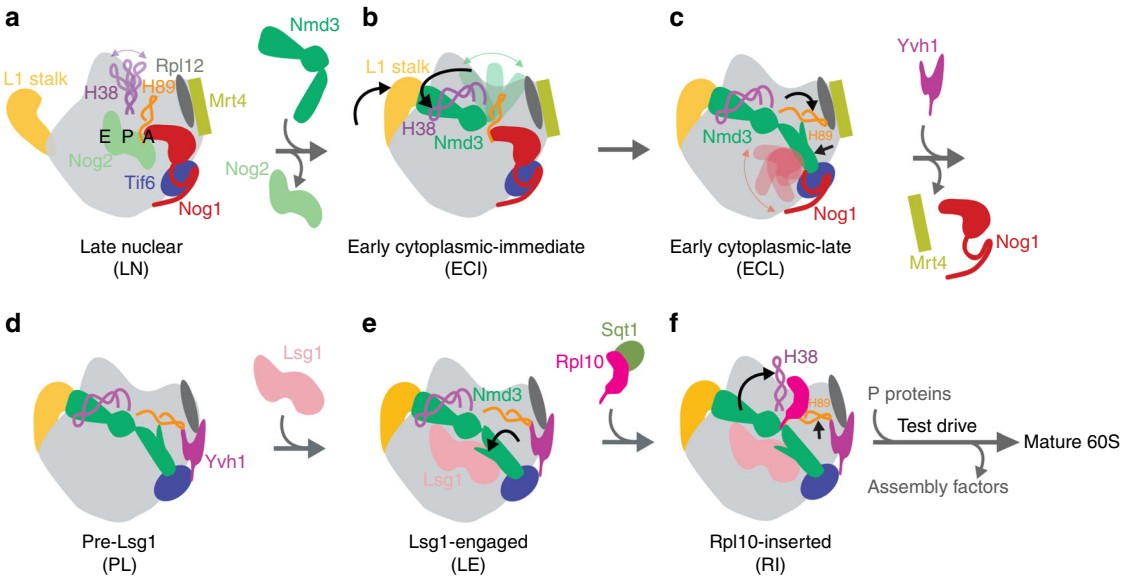

**Fig. 5** Summary of 60S maturation concluding with Rpl10 (uL16) insertion. **a** Late nuclear particle before Nmd3 loading. H38 tip is flexible while H89 is displaced by the N-terminal domain (NTD) of Nog1 in the A site. Rpl12 (uL11) and Mrt4 both bind to the P stalk. **b** Early cytoplasmic-immediate particle showing that the nuclear export factor Nmd3 closes the L1 stalk and captures H38. The NTD of Nog1 remains in the A site, displacing H89, and the NTD of Nmd3 is not discernible, probably because of its high mobility before docking onto Tif6. **c** In the early cytoplasmic-late particle, the NTD and GTPase domain of Nog1 have been released from the A site, allowing Nmd3 to dock on Tif6 and bind H89, which is rearranged to its near-mature position. Note that the Nog1 C terminus remains in place while the NTD and GTPase domain are not discernible, likely due to high flexibility after displacement from the A site. **d** In the pre-Lsg1 particle, multiple assembly factors have been released. The release of Nog1 allows the binding of Yvh1 to release Mrt4 from the P stalk. **e** The Lsg1-engaged particle reveals a rotation of the NTD of Nmd3 away from H89 to engage with Lsg1. This releases H89 from Nmd3 to prime the subunit for the insertion of Rpl10. **f** The insertion of Rpl10 causes retraction of both H38 and H89 to their mature positions to stabilize Rpl10 in its binding site. The complete release of Nmd3 from H38 and H89 poises Nmd3 for its imminent release. Subsequently, the nascent 60S subunit undergoes a test drive using molecular mimics of translation factors before licensing it for bona fide translation

remains in place, but the NTD and GTPase domains have disengaged from the joining face to allow Nmd3 to dock on Tif6 (Fig. 5c). Thus, the release of Nog1 from the joining face can be uncoupled from its release from the exit tunnel. The mechanism for activation of the GTPase activity of Nog1 to allow Nmd3 docking is unclear, as there are no obvious structural changes or protein exchanges in the vicinity of the GTPase domain, with the exception for the N terminus of Nmd3 docking on Tif6. It may be the N terminus of Nmd3, itself, that activates Nog1 to release the NTD from the A site.

The release of Nog1 allows H89 to rearrange into its nearly mature conformation in the A site, promoted by engagement with the NTD of Nmd3 through its histidine thumb. In this state, Nmd3 holds open the Rpl10 binding site through its interactions with H38 and H89, the two helices that form the binding cleft for Rpl10. However, Rpl10 insertion appears to require release of H89 from Nmd3, which is promoted by rotation of the histidine thumb away from H89 to engage the GTPase Lsg1 (Fig. 5d, LE particle). Release of H89 allows it to reposition upon Rpl10 loading to stabilize Rpl10 in its binding site (Fig. 5e, RI particle). Further stabilization comes from the release of H38. Our mechanistic interpretation of Rpl10 loading, based on our structures, is well supported by long-standing genetic interactions between *NMD3* and *RPL10*. We show that mutations in Nmd3 that weaken its affinity for 60S subunits[28] map to its contacts with H38 and H89. These mutations suppress a temperature-sensitive mutation in Rpl10, facilitating its assembly into ribosomes by releasing helices 38 and 89 to stabilize the temperature-sensitive mutant protein. Previously, we showed that these same mutations promote the release of Nmd3 from subunits that contain the mutant Rpl10. These results reveal the intricate interplay between Nmd3 and Rpl10, where

Nmd3 facilitates the loading of Rpl10, which in turn is necessary for the release of Nmd3. The insertion of Rpl10 completes the PTC by providing an unstructured loop that interacts with and stabilizes P site ligands. The P site loop of Rpl10 is necessary for the release of Nmd3. Because the P site loop of Rpl10 was not resolved in our structures, how the P site loop drives Nmd3 release remains an open question.

The P stalk engages with and activates the GTPases that drive the translation cycle. We and others previously identified Yvh1 as a biogenesis factor that promotes P stalk assembly by displacing the placeholder protein Mrt4[26,27]. However, the structure of Yvh1 and its mode of action were unknown. Here, we identified Yvh1 as an integral protein situated between Tif6 and the P stalk in particles affinity purified with Nmd3-TAP (Fig. 5d–f). The density of Yvh1 was present on previous structures[50]; however, it was not attributed to any biogenesis factor. Here, we unambiguously identify this density as Yvh1. Based on comparison of our Mrt4-bound and Yvh1-bound pre-60S complexes, we propose that Yvh1 releases Mrt4 through an allosteric change in its binding site. High-affinity binding of Mrt4 to ribosomal RNA (rRNA) involves A1221 on H42, which fits into a small pocket in Mrt4. Binding of Yvh1 leads to a rotation of the H43 and H44 RNA on the P stalk, forcing Mrt4 to move away from A1221 and destabilizing its interaction with the P stalk.

It has been proposed that Yvh1 is needed for recruitment of the Mex67-Mtr2 heterodimer to helices 42 and 43 of the P stalk for nuclear export of pre-60S[50]. However, our results show that Yvh1 is blocked from joining the pre-60S particle until Nog1 is released, downstream of the activity of Drg1, which initiates cytoplasmic maturation. The inhibition of Yvh1 binding by Nog1 is also supported by recent quantitative mass spectrometry and genetics[23]. Thus, our structural analysis does not support the inference that

Yvh1 promotes export by facilitating the loading of the Mex67-Mtr2 dimer.

Our structural analysis follows assembly of the 60S subunit through insertion of Rpl10 to complete the PTC. At this stage, the subunit is poised for addition of the P stalk to recruit factors for the test drive before being licensed for translation[33,34]. Completion of the PTC by the insertion of the P site loop of Rpl10 and retraction of Nmd3 from the P site allow Sdo1 to probe the integrity of the P site and activate the eEF2-like GTPase Efl1 to release Tif6[34,35,51]. Like translational GTPases, Efl1 is thought to monitor multiple elements of the 60S subunit for correct assembly, including the P stalk, H89, the PTC, and the SRL. Because Tif6 inhibits the engagement of the 60S subunit with the small subunit[19,20], it is the release of Tif6 by Efl1 that licenses the subunit for translation.

## Methods

**Cell growth**. All cells were grown at 30 °C in appropriate dropout medium supplemented with 2% glucose or 1% galactose as the carbon source. Site-directed mutations in *NMD3* were generated by inverse PCR in the vector pAJ123. Suppression of *rpl10-G161D* was tested by transforming plasmids into AJY1657 and plating on Leu-deficient media with glucose.

**Affinity purification of the Rlp24ΔC-TAP particles**. A cell culture of strain AJY1134 (with mutation reg1-501 that eliminates glucose repression of GAL genes) containing pAJ3965 was grown to $OD_{600}$ of 0.3 in 1.5 L of Ura-deficient media with glucose. Galactose was added to a final 1% (w/v) concentration and cells were grown for additional 2 h, harvested, and frozen. Cell pellets were stored at −80 °C. Approximately one third of the cell pellet was used per affinity purification. Cell pellets were washed and resuspended in 1.5 volumes of lysis buffer (20 mM HEPES, pH 7.5, 10 mM $MgCl_2$, 100 mM KCl, 5 mM β-mercaptoethanol, 1 mM PMSF, and protease inhibitors). Extracts were prepared by glass bead lysis and clarified by centrifugation at 4 °C for 15 min at $18,000 \times g$. NP-40 was added to a final concentration of 0.15% (v/v) to the clarified extract which was then incubated with rabbit IgG (Sigma) coupled Dynabeads (Invitrogen) for 1 h at 4 °C. The Dynabeads were prepared as previously described[52]. Beads were then washed once with lysis buffer containing 0.15% NP-40 and 1 mM dithiothreitol instead of β-mercaptoethanol at 4 °C for 5 min, and twice with the same buffer but without NP-40. The bound complexes were enzymatically eluted with tobacco etch virus protease for 90 min at 16 °C. Eluted particles were flash frozen in 10 μL aliquots in liquid $N_2$ and then stored at −80 °C for cryo-EM.

**Purification of particles from diazaborine-treated cells**. Cells with C-terminally TAP-tagged Nmd3 (AJY1874) were grown in 1 L of YPD media to $OD_{600}$ of 0.4. Diazaborine (Millipore) dissolved in dimethyl sulfoxide was added to the cell culture to obtain a final concentration of 10 μg/mL and cultures were shaken at 30 °C for 30 min. Diazaborine-treated cell cultures were divided into three equal parts and cells were harvested by centrifugation. Cell pellets were stored at −80 °C without washing until further use. For purification of diazaborine-treated particles, cell pellets were washed in buffer without diazaborine and particles were purified as described above.

**Cryo-EM grid preparation and data collection**. NC-flat holy carbon grids (CF-1.2/1.3, Protochips Inc.) were pre-coated with a thin layer of freshly prepared carbon film and glow-discharged for 30 s using a Gatan Solarus plasma cleaner before addition of sample. Then, 2.5 μL of affinity-purified particles (~50 nM for the Rlp24ΔC-TAP particles and ~100 nM for the Nmd3-TAP particles isolated from diazaborine-treated cells) were placed onto grids, blotted for 3 s with a blotting force of 5, and rapidly plunged into liquid ethane using a FEI Vitrobot MarkIV operated at 4 °C and 100% humidity. Data were acquired using a FEI Titan Krios transmission electron microscope (Sauer Structural Biology Laboratory, University of Texas at Austin) operating at 300 keV at a nominal magnification of ×22,500 (1.1 Å pixel size) with defocus ranging from −1.0 to −2.5 μm. The data were collected using a total exposure of 6 s fractionated into 20 frames (300 ms per frame) with a dose rate of ~8 electrons per pixel per second and a total exposure dose of ~40 $e^-/Å^2$. A total of 7403 and 6179 micrographs of the Rlp24ΔC-TAP particles and the Nmd3-TAP particles purified from diazaborine-treated cells, respectively, were automatically recorded on a Gatan K2 Summit direct electron detector operated in counting mode using the MSI-Template application within the automated macromolecular microscopy software LEGINON[53].

**Cryo-EM data processing**. All image pre-processing was performed in Appion[54]. Individual movie frames were aligned and averaged using 'MotionCor2' drift-correction software[55]. These drift-corrected micrographs were binned by 8, and bad micrographs and/or regions of micrographs were removed using the 'manual masking' command within Appion. A total of 326,567 and 485,448 particles of the Rlp24ΔC-TAP particles and the Nmd3-TAP particles from diazaborine-treated cells were picked with a template-based particle picker using a reference-free two-dimensional (2D) class average from a small subset of manually picked particles as templates, respectively. The contrast transfer function (CTF) of each micrograph was estimated using CTFFIND4[56]. Selected particles were extracted from micrographs using particle extraction within RELION[57] and the EMAN2 coordinates exported from Appion.

The workflow for data processing is summarized in Supplementary Figs 2 and 4. Briefly, between two and three rounds of reference-free 2D classification with 100 classes for each sample were performed in RELION to remove junk particles, resulting in a clean stack of 216,030 particles for the Rlp24ΔC-TAP sample purified from yeast cells and 393,665 particles for the Nmd3-TAP sample purified from diazaborine-treated yeast cells. Using a previously published pre-60S structure (EMD-8368)[17] low-pass filtered to 60 Å as a reference, three-dimensional (3D) classification was performed within RELION to further sort the particles. The Rlp24ΔC-TAP particles were subjected to two rounds of 3D classification and an additional round of 2D classification, resulting in 178,310 'good' particles for further processing. The Nmd3-TAP particles were subjected to one round of 3D classification, resulting in 340,712 'good' particles for the final processing steps. Next, 3D auto-refine was performed in RELION using the best corresponding model from 3D classification low-pass filtered to 60 Å as a starting model to produce maps at 3.8 Å and 3.7 Å resolution (using the 0.143 gold-standard Fourier shell correlation (FSC) criterion) for the Rlp24ΔC-TAP and the Nmd3-TAP particles, respectively. In order to improve the resolution, movie refinement was also performed using RELION[58] with a running average window of 3 movie frames and a standard deviation of 1 pixel for the translational alignment, followed by 'particle polishing'[59] in RELION to correct for beam-induced motion of individual groups of particles (200 pixel standard deviation) and to perform B-factor weighting using a running average window of 3 frames. These polished particles were then used for another round of 3D auto-refine, resulting in reconstructions at 3.6 Å resolution (again using the gold-standard FSC from two independent half-maps) for both the Rlp24ΔC-TAP and the Nmd3-TAP particles. To improve the local density of important regions of the map and to separate various states, a series of focused 3D classification and signal subtraction using masked regions were performed on the two data sets (Figs. S2 and S5), followed by 3D auto-refine of selected particles. Finally, selected 3D auto-refine maps were subjected to post-processing within RELION with a soft mask and a B-factor of −40 $Å^2$. The overall resolution of each structure was calculated based on the gold-standard 0.143 FSC criterion using two independent half-maps. The local resolution was also estimated using RELION (Supplementary Fig. 1 and 3). The final reconstructions were segmented using Segger[60] in UCSF (University of California, San Francisco) Chimera[61].

**Atomic model building and refinement**. The majority of the model building was performed by manually adjusting and extending existing atomic structures, which had been rigid-body docked into our EM density in Chimera. The atomic structure of a nuclear pre-60S intermediate (PDB 3JCT)[24] was used as the initial model for modeling the LN, ECI and ECL structures with Rpl1, Rpl29 and Rpl42 from PDB 5T62 and Rpl12 from PDB 4V88, and part of the rRNAs from PDB 5T62[17] and PDB 4V88[62]. Rpl12 is not modeled in the ECI and ECL structures due to poor density. The atomic structure of a cytoplasmic pre-60S particle (PDB 5T62)[17] was used as the initial model for modeling the PL, LE, and RI structures with Rpl12 and part of the rRNAs from PDB 4V88[62]. The structures were optimized by manually inspecting and adjusting the RNA backbones/bases and side chains of amino acids into the corresponding cryo-EM density in Coot[63]. Yvh1 was built completely de novo by tracing the chain first and then using large side chains to register the primary sequence within the map. The zinc-binding domains were also used as anchors and sanity checks for our modeling. The final atomic model was refined with real-space refinement[64] in PHENIX[65], and the refinement statistics are shown in Supplementary Table 1. All structural analysis and figure preparation were performed with UCSF Chimera, Coot, and PyMOL[66].

**Reporting summary**. Further information on experimental design is available in the Nature Research Reporting Summary linked to this article.

## Data availability

The data that support the findings of this study are available from the corresponding author upon request. The cryo-EM structures of LN, ECI, ECL, PL, LE and RI pre-60S intermediates have been deposited into the Electron Microscopy Data Bank with accession numbers EMD-0369, EMD-0370, EMD-0371, EMD-0372, EMD-0373, and EMD-0374, respectively. Their associated atomic models have been deposited into the Protein Data Bank with PDB codes 6N8J, 6N8K, 6N8L, 6N8M, 6N8N, and 6N8O, respectively.

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

## Acknowledgements

We thank J. Huibregtse and A. Matouschek for critical reading of the manuscript; A. Dai for help with data collection and processing; D. Wrapp, M. Liu, N. Wang, M. Gilman, and J. McLellan for support with model building; J. Yelland for help with data collection; and members of the Johnson and Taylor labs for helpful discussions. EM data were acquired at the Sauer Structural Biology Laboratory at UT Austin. This work was supported in part by Welch Foundation Grant F-1938 (to D.W.T.) and NIH Grants GM53655 and GM127127 (to A.W.J.). D.W.T. is a CPRIT Scholar supported by the Cancer Prevention and Research Institute of Texas (RR160088).

## Author contributions

Y.Z. performed electron microscopy, single-particle processing, and model building. S.M. purified the pre-60S samples and performed genetic studies. Y.Z., S.M., A.W.J. and D.W.T. analyzed and interpreted the data and wrote the manuscript. A.W.J. and D.W.T. supervised the study and secured funding for the work.

## Additional information

**Competing interests:** The authors declare no competing interests.

**Journal Peer Review Information**: *Nature Communications* thanks Vikram Panse, and the other anonymous reviewer(s) for their contribution to the peer review of this work. Peer reviewer reports are available.

