## [Peer Review File · Nature Communications]

Reviewers' comments:

Reviewer #1 (Remarks to the Author):

In this paper, Zhou et al. present a series of 6 cryo-EM reconstructions of 60S ribosomal biogenesis intermediates derived from 2 different samples in which the function of Drg1, a AAA-ATPase tagged, was modulated. In silico classification of particles obtained from these samples yielded an ensemble of structures representing various late states of 60S biogenesis.

Overall, the data is compelling and describes the sequential structural changes that define the late maturation of the large subunit. The reconstructed models allow the authors to define key RNA rearrangements triggered by changes in protein composition and these structures substantially extend our knowledge of the specific roles of a number of biogenesis factors during 60S maturation in the cytoplasm.

Importantly, they show that manipulation of specific steps in the pathway is critical to trap transient intermediates, as attempts to isolate similar intermediates from steady state samples have not revealed a similar range of structures.

I recommend that the paper be accepted for publication without changes.

Reviewer #2 (Remarks to the Author):

The authors report a serial of pre-60S structures before and after nuclear export, providing several pieces of important structural and mechanistic information that are lacking in the current literatures for final steps of ribosome assembly in cytoplasm. While the overall quality of the manuscript is high, a few issues need to be addressed and clarified. The authors could increase the length of the manuscript to provide more details of structural analysis and interpretation.

(1) One major concern is the interpretation of structural intermediates reported in the manuscript. The authors used mutant cells (or small molecular disrupted cells) to isolate assembly intermediates and tried to place the structures into a linear (sequential) assembly pathway. However, ribosome assembly is known to take parallel pathways and minor populations on assembly branches might accumulate in mutant cells. For example, earlier published work has shown that ITS2-containing pre-60S ribosomes in disrupted cells could even go into cytoplasmic 80S. The authors should take account of this possibility in the data interpretation.

(2) After Nog1 release, did the authors observe Reh1 in the peptide exit channel?

(3) In the LN state, did the authors see any structural difference with the previous Nog2-containing structures, other than the removal of ITS2?

(4) Does Yvh1 interact with Rpl40? Does Rpl40 play any role in releasing Mrt4?

(5) In Fig3, the authors stated that "upon Lsg1 the NTD of Nmd3 undergoes a ~60-degree rotation upon Lsg1 binding". How does this position of Nmd3 relate to the two previously published structures (Ma et al. 2017 NSMB; Malyutin et al., 2017, EMBO J)?

(6) In ED figure 6, panel a does not contain Nog2.

(7) The last three structures (Fig1d-1f) are from diazaborine-treated cells. This drug is supposed to inhibit the function of Drg1. Did the author see similar structures of Fig. 1a-c in the dataset of diazaborine-treated cells?

Reviewer #3 (Remarks to the Author):

The Johnson and Taylor labs report six structures of the yeast 60S pre-ribosomes ranging from late nuclear to cytoplasmic steps. The presented study now provides a comprehensive view on stepwise conformational changes during pre-60S maturation and points new insights of this process. Using C-terminal truncation of Rlp24, authors managed to purify very late nuclear (LN) particles – later stage then published Arx1 or Nog1 particle (Leidig et al., 2014; Wu et al., 2016) and early cytoplasmic (EC) ones as the cytoplasmic maturation is blocked. Even though the LN particle is not yet engaged with export adaptor Nmd3, authors see density for Rpl12 on this particle, processed ITS2 and 5S rotated into its mature conformation. Based on the presented structures, authors explain molecular mechanism of uL16 (Rpl10) incorporation in the cytoplasm, which involves major rRNA rearrangements triggered by Nmd3 binding. The authors validated their model of Nmd3 opening a binding site for uL16 (Rpl10) by functional data. EC particles were further classified to early cytoplasmic -immediate and early cytoplasmic –late, which differ in Nmd3 engagement with Tif6 and release of N-terminal domain of Nog1 in later mentioned particle.

The authors obtained additional three particles – Pre-Lsg1 (PL), Lsg1-engaged (LE) and Rpl10-inserted (RI) by purification of Nmd3 particle from cells mildly treated with diazaborine drug, which inhibits activity of Drg1, an assembly factor initiating cytoplasmic maturation. Compared to previously published Nmd3 particle (Ma et al., 2017), authors detected additional density for Yvh1 factor already in PL particle after release of Nog1, Mrt4, Rlp24 and Bud20. By comparing Mrt4- and Yvh1-bound particles authors deduce the Yvh1 mechanism of Mrt4 release, which is in agreement with previously published genetic data (Kemmler et al., 2009; Lo et al., 2009).

The data presented here are of high quality. A major finding here is the location of the assembly factor, Yvh1, previously unresolved on the 60S pre-ribosome, which provided insights into the mechanism of ribosomal stalk assembly. We have only minor points for revision process and we recommend the presented work for publication.

Minor points and questions:

- When referring to ribosomal proteins, it would be better for the structural biology community to state the new nomenclature names. For e.g. Rpl10 (uL16) or uL16 (Rpl10).
- The authors write: Surprisingly, the C-terminus of Nog1 remains in place on the ECL particle. Could that be a consequence of using rlp24 Δ C mutant, which cannot be released from the particle? Could you comment on this in the discussion?
- Nog2 is released from pre-60S by its GTPase activity, which was proposed to be stimulated by prior structural rearrangements by AAA-ATPase Rea1 and its cofactor Rsa4 (Matsuo et al., 2014). The earlier Nog2 particle contains Rsa4 cofactor (Wu et al., 2016). The presented LN particle seems to have Rea1 as well as Rsa4 released. Could the authors detect the proposed structural rearrangements in Nog2 particle and LN particle, which would later trigger Nog2 release?
- On a similar note, the Hurt laboratory had proposed that Yvh1 is released in the nucleus (Sarkar et al. 2016). Given the structural data this possibility can be completely ruled. Can the authors comment on this in the discussion.

Reviewers' comments:

Reviewer #1 (Remarks to the Author):

In this paper, Zhou et al. present a series of 6 cryo-EM reconstructions of 60S ribosomal biogenesis intermediates derived from 2 different samples in which the function of Drg1, a AAA-ATPase tagged, was modulated. In silico classification of particles obtained from these samples yielded an ensemble of structures representing various late states of 60S biogenesis.

Overall, the data is compelling and describes the sequential structural changes that define the late maturation of the large subunit. The reconstructed models allow the authors to define key RNA rearrangements triggered by changes in protein composition and these structures substantially extend our knowledge of the specific roles of a number of biogenesis factors during 60S maturation in the cytoplasm.

Importantly, they show that manipulation of specific steps in the pathway is critical to trap transient intermediates, as attempts to isolate similar intermediates from steady state samples have not revealed a similar range of structures.

I recommend that the paper be accepted for publication without changes.

We thank this reviewer for their very positive response to our work.

Reviewer #2 (Remarks to the Author):

The authors report a serial of pre-60S structures before and after nuclear export, providing several pieces of important structural and mechanistic information that are lacking in the current literatures for final steps of ribosome assembly in cytoplasm. While the overall quality of the manuscript is high, a few issues need to be addressed and clarified. The authors could increase the length of the manuscript to provide more details of structural analysis and interpretation.

We thank the reviewer for appreciating the importance and quality of the work.

(1) One major concern is the interpretation of structural intermediates reported in the manuscript. The authors used mutant cells (or small molecular disrupted cells) to isolate assembly intermediates and tried to place the structures into a linear (sequential) assembly pathway. However, ribosome assembly is known to take parallel pathways and minor populations on assembly branches might accumulate in mutant cells. For example, earlier published work has shown that ITS2-containing pre-60S ribosomes in disrupted cells could even go into cytoplasmic 80S. The authors should take account of this possibility in the data interpretation.

We understand the concern of this reviewer. The 60S assembly pathway is indeed complex. Previously, we have shown that assembly events at the exit tunnel occur in parallel with stalk assembly; however, both pathways converge at the Efl1-dependent release of Tif6. Consequently, we focused our attention on events on the joining face and P-stalk and did not attempt to reconstruct factors at the exit tunnel, as densities at

the exit tunnel were heterogeneous because of our classification approach. In our classification scheme of cytoplasmic particles affinity purified with Nmd3, we discarded one class in which Rpl12 was depleted. We suspect that these particles represent off pathway intermediates, which are assembled and exported without Rpl12 because of the block in Mrt4 recycling. This conclusion is consistent with our previous work (Lo et al 2009 JCB 186:849-862), where we argued that Rpl12 was unstable and easily lost from particles in the absence of Mrt4.

(2) After Nog1 release, did the authors observe Reh1 in the peptide exit channel?

As noted in response to question 1, we did not focus on factors at the exit tunnel as their densities were heterogeneous in our structures, which were classified according to factors on the joining face. Consequently, we did not try to resolve Rei1 or Reh1.

(3) In the LN state, did the authors see any structural difference with the previous Nog2-containing structures, other than the removal of ITS2?

In the text, we noted several structural differences between the Nog2-particle and our LN particle, including the rotation of 5S into its mature orientation and appearance of Rpl29 and Rpl42. In addition, other factors, including Rpf2, Rrs1, Rsa4, Nug1, Nsa2 and Cgr1 are absent from the LN particle, indicating that these factors have been released. Also see response to Reviewer 3 regarding activation of Nog2, which is now included in the revised Discussion.

(4) Does Yvh1 interact with Rpl40? Does Rpl40 play any role in releasing Mrt4?

Although Yvh1 and Rpl40 are close in close proximity, they do not appear to make any direct contacts. Work on the role of Rpl40 is ongoing, but we feel that it is beyond the scope of the current manuscript.

(5) In Fig3, the authors stated that “upon Lsg1 the NTD of Nmd3 undergoes a ~60-degree rotation upon Lsg1 binding”. How does this position of Nmd3 relate to the two previously published structures (Ma et al. 2017 NSMB; Malyutin et al., 2017, EMBO J)?

The rotated state of the NTD of Nmd3 does appear to correspond to the density map reported from Ning Gao’s lab in Ma et al. However, they were unable to trace chains in this density. In Malyutin et al, we reported a structure in which the NTD of Nmd3 was in the H89 position and not rotated toward Lsg1. In that work, we did observe a population of particles in which Nmd3 was rotated toward Lsg1, but we were unable to clearly resolve this conformation.

(6) In ED figure 6, panel a does not contain Nog2.

In the revised manuscript, we now include Nog2 in Supplementary Fig 9, and we discuss the possible mechanism of activation on pg 6, first full paragraph.

(7) The last three structures (Fig1d-1f) are from diazaborine-treated cells. This drug is supposed to inhibit the function of Drg1. Did the author see similar structures of Fig. 1a-c in the dataset of diazaborine-treated cells?

We thank the reviewer for pointing out this important detail. We noted in the text (pg. 3) that we used a concentration of diazaborine that was only partially inhibitory. We also

did not add diazaborine to our extraction buffer. Consequently, we did not effectively inhibit Drg1, and thus we did not recover particles from our diazaborine-treated sample similar to those obtained with Rlp24 Δ C.

Reviewer #3 (Remarks to the Author):

The Johnson and Taylor labs report six structures of the yeast 60S pre-ribosomes ranging from late nuclear to cytoplasmic steps. The presented study now provides a comprehensive view on stepwise conformational changes during pre-60S maturation and points new insights of this process. Using C-terminal truncation of Rlp24, authors managed to purify very late nuclear (LN) particles – later stage than published Arx1 or Nog1 particle (Leidig et al., 2014; Wu et al., 2016) and early cytoplasmic (EC) ones as the cytoplasmic maturation is blocked. Even though the LN particle is not yet engaged with export adaptor Nmd3, authors see density for Rpl12 on this particle, processed ITS2 and 5S rotated into its mature conformation. Based on the presented structures, authors explain molecular mechanism of uL16 (Rpl10) incorporation in the cytoplasm, which involves major rRNA rearrangements triggered by Nmd3 binding. The authors validated their model of Nmd3 opening a binding site for uL16 (Rpl10) by functional data. EC particles were further classified to early cytoplasmic - immediate and early cytoplasmic –late, which differ in Nmd3 engagement with Tif6 and release of N-terminal domain of Nog1 in later mentioned particle.

The authors obtained additional three particles – Pre-Lsg1 (PL), Lsg1-engaged (LE) and Rpl10-inserted (RI) by purification of Nmd3 particle from cells mildly treated with diazaborine drug, which inhibits activity of Drg1, an assembly factor initiating cytoplasmic maturation. Compared to previously published Nmd3 particle (Ma et al., 2017), authors detected additional density for Yvh1 factor already in PL particle after release of Nog1, Mrt4, Rlp24 and Bud20. By comparing Mrt4- and Yvh1-bound particles authors deduce the Yvh1 mechanism of Mrt4 release, which is in agreement with previously published genetic data (Kemmler et al., 2009; Lo et al., 2009).

The data presented here are of high quality. A major finding here is the location of the assembly factor, Yvh1, previously unresolved on the 60S pre-ribosome, which provided insights into the mechanism of ribosomal stalk assembly. We have only minor points for revision process and we recommend the presented work for publication. We thank the reviewer for appreciating the importance and quality of the work.

Minor points and questions:

- When referring to ribosomal proteins, it would be better for the structural biology community to state the new nomenclature names. For e.g. Rpl10 (uL16) or uL16 (Rpl10).

We agree with the reviewer that it is important to introduce the new nomenclature. It becomes slightly complicated when referring to the genes which have different names from their universal nomenclature for proteins. We now include the universal name

when first introducing each rprotein but continue to refer to the rproteins by their yeast names.

- The authors write: Surprisingly, the C-terminus of Nog1 remains in place on the ECL particle. Could that be a consequence of using rlp24 Δ C mutant, which cannot be released from the particle? Could you comment on this in the discussion?

We thank the reviewer for pointing out this important feature. To address this and other questions raised by reviewers 2 and 3 we have completely rewritten the Discussion. We have now added the following (in blue italics) to the Results section:

Surprisingly, the C-terminus of Nog1 remains in place on the ECL particle, intertwining with Rlp24 and occupying the exit tunnel (Fig. 1c). The persistence of the C-terminus of Nog1 in the exit tunnel likely results from the failure to recruit Drg1 to the Rlp24 Δ C particle and suggests that events on the joining face can be uncoupled from extraction of Nog1 from the exit tunnel by Drg1.

We elaborate on this in the Discussion (pg. 6) as follows:

Upon initial binding of Nmd3 (Supplementary Fig. 9 ECL particle), its eIF5A and eL22-like domains occupy the E and P sites, respectively, where they close the L1 stalk and capture H38. In this state the N-terminus of Nmd3 is sterically blocked from docking on Tif6 by the presence of the N-terminal and GTPase domains of Nog1, which disrupt H89 in the A site. The C-terminus of Nog1, on the other hand, wraps around the back of the subunit and enters the exit tunnel. The release of Nog1 is thought to be coupled to the Drg1-dependent release of Rlp24, presumably by extracting the C-terminus of Nog1 from the exit tunnel^{22,30}. However, in the ECL particle, recruitment of Drg1 is blocked due to truncation of the C-terminus of Rlp24. Consequently, the C-terminus of Nog1 remains in place, but the NTD and GTPase domains have disengaged from the joining face to allow Nmd3 to dock on Tif6 (Supplementary Fig. 9). Thus, the release of Nog1 from the joining face can be uncoupled from its release from the exit tunnel. The mechanism for activation of the GTPase activity of Nog1 to allow Nmd3 docking is unclear, as there are no obvious structural changes or protein exchanges in the vicinity of the GTPase domain, with the exception for the N-terminus of Nmd3 docking on Tif6. It may be the N-terminus of Nmd3, itself, that activates Nog1 to release the NTD from the A site.

- Nog2 is released from pre-60S by its GTPase activity, which was proposed to be stimulated by prior structural rearrangements by AAA-ATPase Rea1 and its cofactor Rsa4 (Matsuo et al., 2014). The earlier Nog2 particle contains Rsa4 cofactor (Wu et al., 2016). The presented LN particle seems to have Rea1 as well as Rsa4 released. Could the authors detect the proposed structural rearrangements in Nog2 particle and LN particle, which would later trigger Nog2 release?

In the revised manuscript, we now include Nog2 in Supplementary Fig 9, and we discuss the possible mechanism of activation (pg. 6):

Nuclear export of the 60S subunit requires the conserved multi-domain export adapter Nmd3^{28,29}. In the late nuclear particle, the loading of Nmd3 is blocked by the GTPase Nog2²⁰. Although we do not know how Nog2 is activated or released from the pre-60S particle, our LN structure reveals that the biogenesis factor Rpf2 has been removed at this stage in assembly. The C-terminus of Rpf2 interdigitates between Nog2 and H69 and stabilizes a loop of Nog2 through a beta-sheet interaction. Removal of Rpf2 appears to destabilize H69 as we were unable to resolve this helix in the LN structure. These observations lead us to speculate that the removal of Rpf2 allows conformational changes in Nog2 and H69 that promote the activation of its GTPase activity and/or its release to allow Nmd3 to bind.

- On a similar note, the Hurt laboratory had proposed that Yvh1 is released in the nucleus (Sarkar et al. 2016). Given the structural data this possibility can be completely ruled. Can the authors comment on this in the discussion.

There is indeed a conflict between our work and conclusions drawn in Sarker et al. We have now included the following paragraph in the Discussion (pg. 7):

It has been proposed that Yvh1 is needed for recruitment of the Mex67-Mtr2 heterodimer to helices 42 and 43 of the P stalk for nuclear export of pre-60S³¹. However, our results show that Yvh1 is blocked from joining the pre-60S particle until Nog1 is released, downstream of the activity of Drg1, which initiates cytoplasmic maturation. The inhibition of Yvh1 binding by Nog1 is also supported by recent quantitative mass spectrometric and genetic results³⁰. Thus, our structural analysis does not support the inference that Yvh1 promotes export by facilitating the loading of the Mex67-Mtr2 dimer.

REVIEWERS' COMMENTS:

Reviewer #2 (Remarks to the Author):

The authors have addressed my concerns. I believe the manuscript is ready for publication.

Reviewer #3 (Remarks to the Author):

The authors addressed all minor concerns and questions that we raised, and have carefully revised their manuscript. Therefore, we suggest to accept the manuscript for publication.